# Effect Steel Fibre Content on the Load-Carrying Capacity of Fibre-Reinforced Concrete Expansion Anchor

**DOI:** 10.3390/ma14247757

**Published:** 2021-12-15

**Authors:** Daniel Dudek, Marta Kadela, Marcin Małek

**Affiliations:** 1Building Research Institute (ITB), Ul. Filtrowa 1, 00-611 Warsaw, Poland; d.dudek@itb.pl; 2Faculty of Civil Engineering and Geodesy, Military University of Technology in Warsaw, Ul. Gen. Sylwestra Kaliskiego 2, 00-908 Warsaw, Poland; marcin.malek@wat.edu.pl

**Keywords:** concrete, fibre reinforcement, expansion anchor, mechanical properties, pull-out test, pull-out strength, cracked concrete substrate, non-cracked concrete substrate

## Abstract

The article presents the pull-out strength tests carried out on M10 expansion anchors in non-cracked and cracked concrete with a crack width c_w_ = 0.30 mm. The breaking loads and the average pull-out strength of anchors in fibre-reinforced concrete substrates were determined. Fibre content ratios of 15, 30 and 50 kg/m^3^ were used. In addition, two different classes of concrete (C20/25 and C50/60) were tested. The addition of steel fibres caused a decrease in the pull-out strength by 5% for non-cracked concrete of C20/25 class and fibre content up to 30 kg/m^3^ and a further 7% for the remaining specified dosage. While for concrete of the C50/60 class, it a decrease in the pull-out strength of up to 20% was observed. For cracked concrete class C20/25 with crack initiation c_w_ = 0.30 mm, the reduction was from 9% to 16% in relation to non-cracked concrete and a maximum of 18% for the fibre content of 50 kg/m^3^. The difference between the tensile load capacity of C50/60 class cracked and non-cracked concrete was lower than 5% and fell within the measurement error.

## 1. Introduction

The rapid development of civilisation requires the involvement of new building materials and techniques to face the challenges of the century to come. In recent years, many studies concerning new generations of concretes have been published, including high-performance concrete (HPC), self-compacting concrete (SCC), eco-efficient concrete, and even self-healing concrete [1,2,3]. One of the most interesting and widely used is fibre-reinforced concretes fibre [4,5,6], including recycled fibre [7,8,9]. The fibre-reinforced concretes are characterised by superior resistance to crack propagation and cracking itself (by fibre bridging on a crack), impact, and fatigue, having good durability at the same time [10]. Moreover, fibre reinforced concretes can particularly improve the stiffness and durability of reinforced and prestressed concrete structures [11]. Due to these properties, fibre-reinforced concrete found its application in the construction of building, bridges, tunnels, and heavy-duty pavement [4]. The steel fibre reinforced concretes (SFRC) are the most widely used [12,13,14,15]. Shi et al. [16] presented the research and evaluated the seismic performance of beam–column joints in steel fibre reinforced high–strength concrete (SFRHC). While Jiang et al. [17] presented the resistance to chloride penetration of the concrete. The results showed that the expansion caused by the hydration of MgO could suppress the shrinkage of the bridge deck, effectively preventing shrinkage cracks.

Nowadays, due to energy efficiency and sustainability, an increase in the use of building insulation is observed [9]. Moreover, civil engineering tends to maximise the use of elevation (solar panels, banners, etc.) what causes a significant increase in the utilisation of anchors. When considering an increase in the application of fibre-reinforced concrete application, the issue of joining building elements to the fibre-reinforced concrete should be considered. In traditional concrete (without fibre), one of the most common solutions is fasteners in the form of expansion anchors placed in concrete drilled holes [18]. The structural integrity of the joint is determined by several factors including applied force (value and type), water content, type of concrete, and its tendency to carbonation [18,19,20,21]. In recent years, many researchers have undertaken an effort to investigate the strength of such joints in terms of other operating conditions, e.g., cracked substrate or additional loads [22,23,24,25,26]. Giresini et al. [27] focused on the issue of experimental pull-out tests and design indications for strength anchors and proposed the uniform stress model to predict pull-out strength values of installed anchors. Ruopp et al. [28] did very extensive research concerned with the behaviour of large anchor plates under shear loading in steel-to-concrete joints and determined the load distribution of forces at the different rows of studs on the anchor plate. Luo et al. [29] investigated steel plate-concrete joints connected with anchors and adhesive in terms of sheer performance and discovered that the ultimate loads of the anchor- and adhesive-connected specimens were approximately 78% of the sum of adhesive-connected and anchor-connected specimens.

Anchoring in the fibre reinforced concrete was tested by Lee et al. [30], Bokor et al. [31] and Gesoglu et al. [32]. Lee et al. [30] performed the research on shear failure and concrete edge breakout resistance of anchors in SFRC and reported that the shear load and displacement capacities of the SFRC anchors increased linearly with the increase in the fibre volume fraction of fibres. Bokor et al. [31] investigated fasteners in SFRC subjected to increased loading rates and showed the beneficial effect of the steel fibres on concrete cone capacity of anchors, increasing its value by 22% and 37% for 30 kg/m^3^ and 50 kg/m^3^ fibre content respectively. Gesoglu et al. [32] researched the tensile behaviour of post-installed anchors in SFRC and stated that the anchors’ pullout capacities were not significantly affected by the addition of steel fibres but there was a reduction of damage on concrete compared to the plain-one. The small amount of research on this issue and significant discrepancies between the results of different scientists indicated a large gap in the current state of the knowledge. Currently, fastenings with steel anchors are mainly performed in concrete foundations, and their technical assessment is based on the recommendations of EOTA [33] and Eurocode 2 [34]. Fastenings made in fibre-reinforced concrete elements have so far been designed and implemented as for typical concrete substrates. It is not consistent with the actual working conditions of the connectors in structural fastenings. However, the rules for making structural fastenings in fibre-reinforced concrete substrates and for carrying out their technical assessments have not been developed so far. Therefore, this paper aims to affect the steel fibres content on the maximum pull-out force for M10 expansion fasteners installed in fibre-reinforced concrete substrates. The article presents the pull-out strength tests carried out on M10 expansion anchors in non-cracked and cracked steel fibre reinforced concrete. Two concrete classes (C20/25 and C50/60) were tested to determine the possibility of such fastener applications in civil engineering. Unlike other scientists, pull-out tests were carried out after 90 days, which aims to approximate the actual working conditions [35,36] and start research on the influence of the increase in the compressive strength of the fibre-reinforced concrete substrate over time on the pull-out test results. Such research has not been conducted so far. The obtained results are the first step to change in the procedure of technical assessment for steel expansion anchors in fibre-reinforced concrete substrate and taking into account the fibre content in designing the application of the anchors in fibre-reinforced concrete substrate.

## 2. Materials

### 2.1. Steel Anchors

This study used M10 torque-controlled anchors LE-ZNA4 (Klimas, Mykanow, Poland). The product has a valid European Technical Assessment. The geometry (Figure 1) and tensile strength of the used expansion anchor are given in Table 1. Installation parameters with material data in accordance with guidelines are given in Table 2.

### 2.2. Concrete Base

#### 2.2.1. Products of Concrete Base material

The materials used in this study were Portland cement (Górażdże Cement S.A., Chorula, Poland), steel fibres and tap water. The industrial Portland cement was CEM I 42.5R, according to PN-EN 197-1:2012 [37]. Its chemical composition measured as per PN-EN 196-2:2013-11 [38] (LOI—loss on ignition; IR—insoluble residue) and physical properties measured according to PN-EN 196-6:2011 [39] are given in Table 3 and Table 4 respectively. The compressive strength of cement was determined according to PN-EN 196-1:2016-07 [40].

Steelbet 50/0.8 (Urban-Metal, Poland) steel fibres were used. The fibres are made of low carbon steel (Figure 2). The measured physical properties are given in Table 5.

#### 2.2.2. Concrete Substrate

The concrete bases were prepared using the formula and aggregate grading curve prescribed in EADs [33] and Eurocodes [34] for steel anchors. Four mixtures for each concrete class C20/25 and C50/60 were produced (three different contents for fibre and base–mixture without fibre). The mixture was prepared in a concrete factory and delivered to the laboratory of Building Research Institute, where the concrete substrate was made. The composition of the concrete mixtures of class C20/25 and C50/60 are given in Table 6. Fibre content ratios of 15, 30 and 50 kg/m^3^ were used. So, steel fibre contents were 6.5, 13.0 and 21.5% of cement, respectively, for concrete C20/25 and 3.6, 7.2 and 11.5% of cement weight, respectively, for concrete C50/60.

The fibres were dosed manually to the concrete mixture, after mixing other components for 15 min. The whole mixing time lasted for a maximum of 20 min. The mixture was poured in a rectangular mould of a concrete substrate with dimensions 4000 × 1000 × 400 mm dimensions. The concrete mixtures were compacted in the mould using a hand vibrator (GEKO, Radomsko, Poland). All samples were obtained and stored in laboratory conditions (21 °C temperature and 50% humidity). Before the test itself, the anchors were installed in drill holes with *d_cut_* as nominal.

#### 2.2.3. Concrete Samples

While preparing the concrete substrates, a part of the mixture was poured into a 150 mm × 150 mm × 150 mm mould to determine the compressive strength. In this case, the concrete mixture was compacted using a compactor. All samples were obtained in laboratory conditions (21 °C temperature and 50% humidity). The samples for testing the properties of the substrate were stored in water according to PN-EN 12390-2:2019 [41].

## 3. Methodology

### 3.1. Test Procedure

In order to conduct the research, the experiments were carried out in the following stages:(1)Concrete substrates with different fibre content were prepared; the description was shown in Section 2.2.2;(2)While preparing the concrete substrates, the samples are taken for compressive strength testing, see Section 2.2.3;(3)The compressive strength tests of concrete samples with different fibre content were carried out after 28 days (to determine 28-days compressive strength) and 90 days (to compare compressive strength with results of pull-out tests) according to the procedure described in Section 3.2;(4)The pull-out tests were carried out after 90 days, see Section 3.4;(5)After the pull-out tests, the boreholes were drilled from the concrete substrate and the actual fibre content of the concrete was determined according to the procedure presented in Section 3.3.

### 3.2. Compressive Strength of Concrete Sample

Compressive strength was measured on samples 150 mm × 150 mm × 150 mm, according to PN-EN 12390-3:2011 + AC:2012 [42], using a compression machine MEGA 6-3000-100 (FORM + TEST, Riedlingen, Germany) having 3000 kN maximum load capacity. The tests were carried out on samples after 28 and 90 days of curing. The test after 90 days was carried out due to the possibility of comparing the results of compressive strength for concretes without fibres and reinforced with steel fibres when analysing the results of the pull-out test, which is performed after 90 days, see Section 3.3. Three samples per mix were used.

### 3.3. Steel Fibre Content in Concrete

The fibre content in concrete was determined on samples 109 mm × 109 mm, according to PN-EN 14721 + A1:2007 [43], using an electronic scale WTC 3000 (RADWAG, Radom, Poland) with a maximum weight of 3100 g. The tests were carried out on samples taken from boreholes in concrete substrates of class C20/25 and C50/60 with an addition of steel fibres. The boreholes and crushed samples are shown in Figure 3.

Three samples per mix of fibre-reinforced concrete were tested.

### 3.4. The Pull-Out Test

The tests consisted in determining the maximum pull-out force for steel expansion anchors in C20/25 and C50/60 concrete substrate (with and without fibre). Without accounting for spacing the distance of the anchors from the edge and the thickness of the base, pull-out strength determined based on tests of steel expansion anchors subjected to static or quasi-static load is described by the Formula (1).
(1)$$F_{Ru,cone}=F_{Ru}^{t}\cdot {\left(\frac{f_{c}}{f_{c,test}}\right)}^{0.5}$$

The tests were performed in a special test station enabling the initiation of cracks with the width of 0.30 mm (Figure 4), using a force sensor C6 A (HBM, Darmstadt, Germany) with a maximum force of 200 kN, displacement sensors WA50 (HBM, Darmstadt, Germany) with a maximum displacement of 50 mm, Zwick (Zwick, Ulm, Germany) own configuration devices Different concrete bases were prepared for the test–non- and cracked with crack width of *c_w_ =* 0.30 mm. The method of creating a crack is described in the guidelines for steel expansion fasteners.

The pull-out tests were carried out 90 days after forming the substrates. The tests should be carried out so that the pulling out was not affected by other external factors, such as temperature, humidity and distances from the edge and axis of the anchors.

## 4. Results and Discussion

### 4.1. Compressive Strength

The results of the compressive strength tests of concrete samples for fibre-reinforced concrete of the C20/25 class are shown in Figure 5. A decrease in compressive strength was observed depending on the fibre content in the concrete mixture. The 28-days compressive strength was characterised by a decrease in strength by 15%, 27% and 30% compared to the base sample for the content of 15, 30 and 50 kg/m^3^, respectively. The decrease in compressive strength results from the reduced aggregate content in the concrete mixture.

In each case, the 90-days compressive strength was higher than after 28 days. After 90 days compressive strength for concrete without and with steel fibre addition increased by an average of 12% compared to 28 days compressive strength. However, the decrease in strength after 90 days was the same as for 28 days depending on the fibre content in the concrete mixture.

The compressive strength of samples for C50/60 class concrete is shown in Figure 6. Additionally, for this class, the analogy changes in compressive strength were observed depending on the test time and the content of steel fibres. It can be observed that the assumed compressive strength was achieved after 28 days and the further changes in the strength of the concrete substrate for all mixtures were slight.

The maximum decrease in 28 and 90 days compressive strength were 12% and 17% respectively. Thus, these are definitely smaller decreases in strength compared to the C20/25 class concrete due to the amount of aggregate and the addition of steel fibres and its bulk density.

These observations of a decrease in compressive strength with steel fibre addition are in line with the results of other scientists. Lehner et al. [15] obtained lower resistance of fibre-reinforced concrete for higher fibre content. In literature, the decrease or increase in compressive strength with the addition of fibres was observed. Ding et al. [44] reported that steel fibre had no significant impact on the compressive strength of hardened concrete. Serrano et al. [45] determined that the compressive strength increased up to 1.0% addition of fibres and then decreased. Other scientists obtained the same conclusion [46,47,48]. Revathi and Kumar determined that compressive strength increased up to 1.2% steel fibre addition, and up to 6% scrap steel fibre addition [46]. While Murali et al. [49] and Song and Hwang [50] determined that the addition of 1.5% showed a significant increase in compressive strength. Marcalikova et al. [51] tested concrete with 0, 40, 75 and 110 kg/m^3^ and obtained maximum compressive strength for steel fibre of 55 kg/m^3^. The properties of reinforced concrete are influenced by the type of fibres, their properties, such as length, diameter proportions and, above all, their content [52].

### 4.2. Steel Fibre Content in a Concrete Mixture

Figure 7 present the real content of steel fibres in the prepared substrates of class C20/25 and C50/60 respectively. It was observed that for concrete C20/25, the real fibre content was 14.7 kg/m^3^, 27.5 kg/m^3^ and 46.0 kg/m^3^, which corresponded to a 2% lower fibre content compared to the assumed content for 15.0 kg/m^3^ and 8% for other fibre content. While for concrete class C50/60, the real content of steel fibres was higher than assumed. The 8% increase in fibres content for the concrete with the addition of fibre above 30.0 kg/m^3^ was determined. Thus, the maximum changes in the fibre content within ±10% can be observed, which is a slight change and is not considered later in the manuscript. The changes in fibre content depend on the method of preparing the concrete substrates and the distribution of the fibres in the concrete mixture. In this study it is directly related to manual dosing, see Section 2.2.2. This is a problem with any mix produced commercially and made on construction sites.

For each case, the lack of fibre agglomeration in the concrete was observed, which is a major problem in fibre reinforced concrete mixtures.

### 4.3. The Results of Pull-out Test

Figure 8, Figure 9, Figure 10, Figure 11, Figure 12 and Figure 13 present the results of pull-out tests carried out 90 days after forming the concrete substrates. The 90 days compressive strength of the base and reinforced concrete was about ±10% compared to 28 days compressive strength, see Section 4.1.

The load capacities were equal to 16.8 MPa for base substrate (without fibre) and were lower for steel fibre-reinforced concrete. For the concrete class of C20/25, a reduction of the load capacity by 5% for the fibre content of 15.0 and 30 kg/m^3^ and 12% for the content of 50.0 kg/m^3^ was obtained (Figure 8). While Farhat et al. [53] obtained load capacity equal to 17.58 kN for the same concrete class (C20/25) and fibre content of 150 kg/m^3^, but they used torque-controlled bolt anchor. Ahmed and Braimah determined that the tensile capacity for undercut anchors installed in concrete with a strength of 95 MPa reinforced with a fibre content of 40 kg/m^3^ increased for installation depth was up to 120 mm and then decreased.

In the next step, the coefficient resulting from the compressive strength of the substrate has been included according to the procedure for determining load capacity in accordance with guidelines.

These were the actual load capacities after the reduction with the coefficient resulting from concrete compression. For non-reduced load capacities, the initial decrease was 13% and 23% for the highest content of steel fibres. Such behaviour of the connectors characterised its structure. Along with the increased fibre content, the connector had the problem of opening the expansion cone when setting the installation torque. In all research cases, the image of destruction was observed as a pull of the connector with the cone. An example of the destruction image is shown in Figure 9.

Similarly, the tests were carried out on substrates where scratching was initiated up to the level of *c_w_* = 0.30 mm (Figure 10). A lower load capacity can be observed than for non-cracked concrete. It is in the line with the observations of other researchers. The same Czarnecki et al. [54] determined reduction in load capacity from 20 to 50% for expansion anchor and unreinforced concrete.

In this study, the decrease in load capacity of fibre-reinforced concrete was in the range of 12 to 18% for the steel fibre content up to 50 kg/m^3^ compared to the cracked substrate from concrete without fibre. The analogy phenomena were determined by Lee et al. [30], but for bonded anchors installed in concrete with a strength of 27 MPa. Due to the specificity of the test stand and the method of scratch initiation, the tear-out cones had different shapes. The reduction in load capacity between non-cracked and cracked concrete was 9% for the substrate without fibres and 15% for the maximum content of fibre content. Thus, in this case, the amount of actual fibre content did not matter in terms of the pull-out resistance of the expansion anchors.

A C50/60 class substrate with the same fibre content was also used for C20/25 concrete. The load capacity for non-cracked concrete of class C50/60 is shown in Figure 11.

A decrease in the pull-out strength by about 5% was observed for the fibre content up to 30 kg/m^3^ and a further 10% for the highest content of fibre dispersion. It can be observed that the highest load capacity for fibre-reinforced concrete was obtained for the fibre content of 30 kg/m^3^. The same trend was observed by Bokor et al. [31] for bonded anchors (M16, 12.9 class) installed in the concrete with a fibre content of 0, 30 and 50 kg/m^3^.

The pull-out resistance was relatively higher than for the C20/25 class substrate and amounts to 25%. It is related to the compressive strength of the concrete substrate and the behaviour of the expansion cone in the structure of the M10 expansion anchors.

A decrease in the pull-out resistance was observed for cracked concrete of class C50/60 with crack initiation *c_w_* = 0.30 mm (Figure 12). The highest decrease was observed for the highest fibre addition, equal to 20%. It can be observed that for the cracked substrate, a decrease in the pull-out load capacity by about 20% compared to non-cracked concrete of the C50/60 class was obtained. For each pull-out performed, cones of average diameter equal to 300 mm were observed. The cones, depending on the amount of steel fibre reinforcement, were characterised by a compact structure for maximum fibre content and a looser/ordinary structure for unreinforced concrete, see Figure 13.

## 5. Summary and Conclusions

The research aimed to assess the possibility of using M10 expansion fasteners installed in concrete substrates of class C20/25 and C50/60 with the addition of steel fibres of various contents. The fibre content of 15 kg/m^3^, 30 kg/m^3^ and 50 kg/m^3^ were used. The pull-out load capacity of steel expansion connectors M10 in non- and cracked (with crack initiation *c_w_* = 0.30 mm) substrate was determined. Based on the results of the experimental study, the following important conclusions can be drawn:(1)There were observed reductions in the compressive strength of C20/25 concrete substrates of the order of 15% for the lowest dose of steel fibres and 30% for the remaining ones, both after 28 days. After 90 days, the increase in strength was 10% compared to C20/25 concrete.(2)For concrete C50/60, there was an increase of 10% after 90 days, and the decreases resulting from the reinforcement were recorded at a maximum of 16%.(3)Steelbet 50/0.8 steel fibres affect the strength of anchors using M10 expansion anchors, reducing pull-out strength.(4)The addition of fibres caused a decrease in the pull-out strength by 5% for non-cracked concrete of C20/25 class and fibre content up to 30 kg/m^3^ and a further 7% for the remaining specified dosage.(5)For cracked concrete with crack initiation *c_w_* = 0.30 mm, the reduction was from 9% to 16% in relation to non-cracked concrete and a maximum of 18% for the fibre content of 50 kg/m^3^.(6)For concrete of the C50/60 class, a decrease in the pull-out strength of up to 20% was observed

The difference between the tensile load capacity of C50/60 class cracked and non-cracked concrete was lower than 5% and fell within the measurement error.

The article is part of a wider research project aimed at assessing the actual load capacity of expansion fasteners in steel fibre reinforced concrete substrates. Moreover, as part of further research, extensive tests of the properties of steel fibre reinforced concrete will be carried out in order to search for the relationship between these parameters and the load capacity of the fasteners. In addition, long-term studies will be conducted to evaluate the long-term effects on the analysed parameters.

## Figures and Tables

**Figure 1 materials-14-07757-f001:**
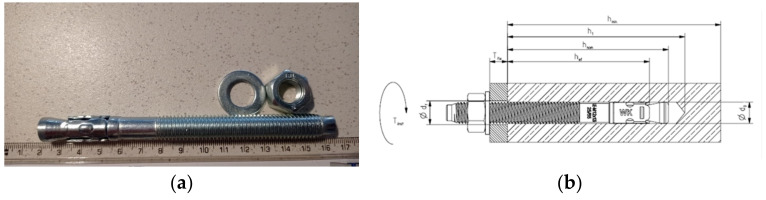
Steel expansion anchor: (**a**) geometry; (**b**) installation parameters.

**Figure 2 materials-14-07757-f002:**
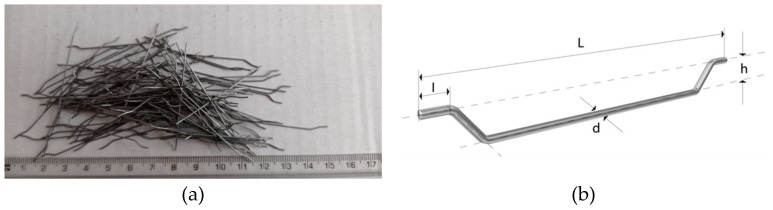
Steel fibres (SF). (**a**) with length 50 mm; (**b**) geometry.

**Figure 3 materials-14-07757-f003:**
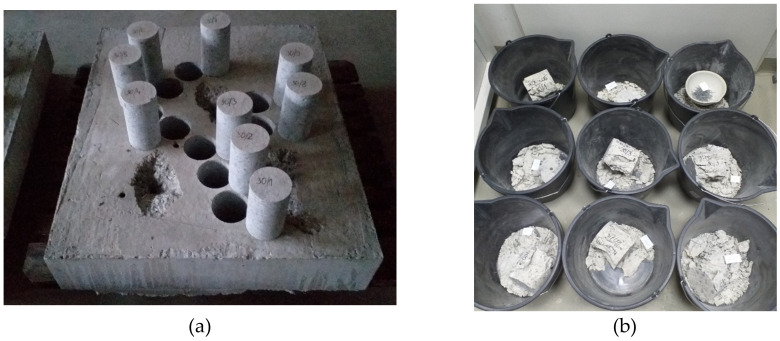
Measuring of steel fibre content: (**a**) sampling, (**b**) tested samples.

**Figure 4 materials-14-07757-f004:**
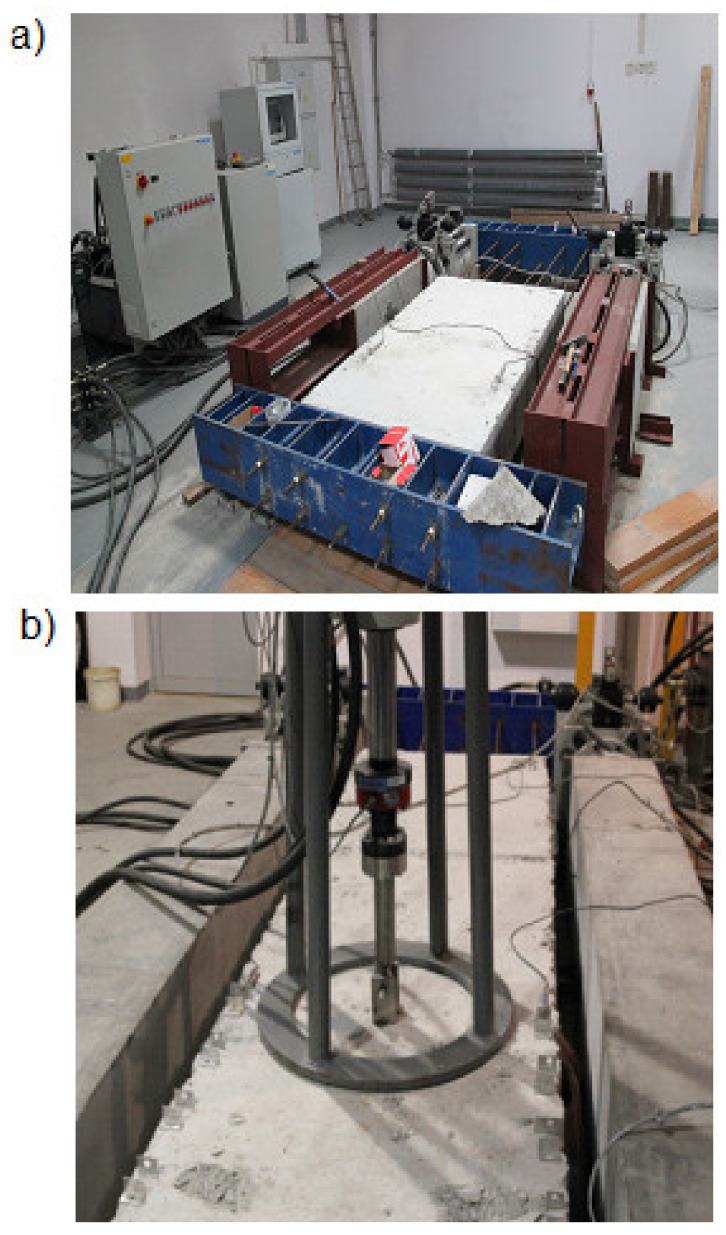
Test station: (**a**) side view, (**b**) view during the tests.

**Figure 5 materials-14-07757-f005:**
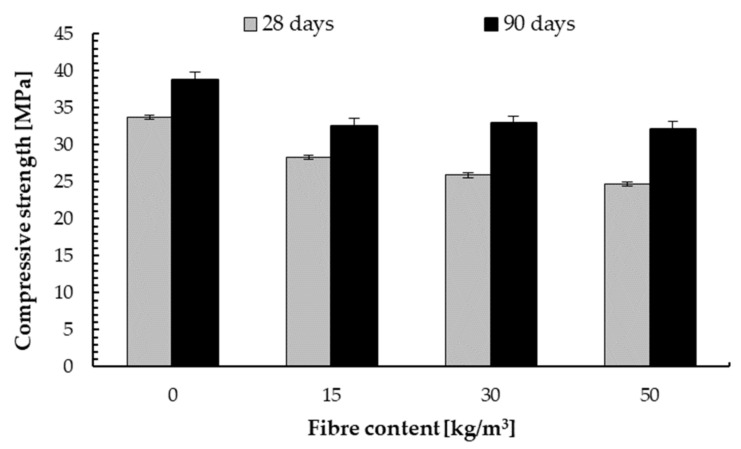
Compressive strength of concrete C20/25 depending on steel fibre content.

**Figure 6 materials-14-07757-f006:**
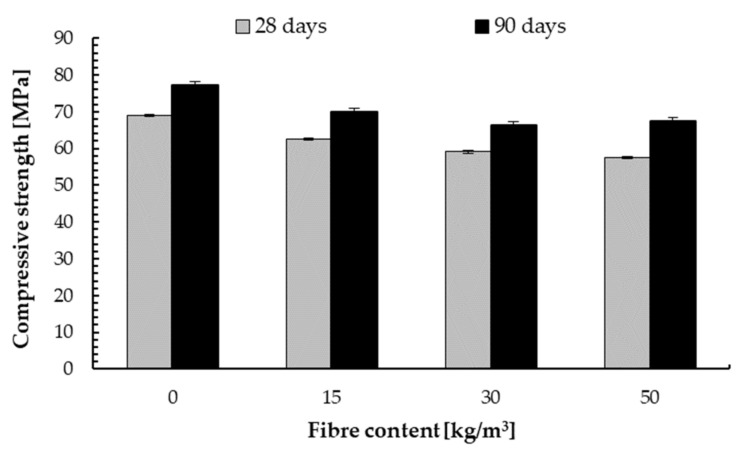
Compressive strength of concrete C50/60 depending on steel fibre content.

**Figure 7 materials-14-07757-f007:**
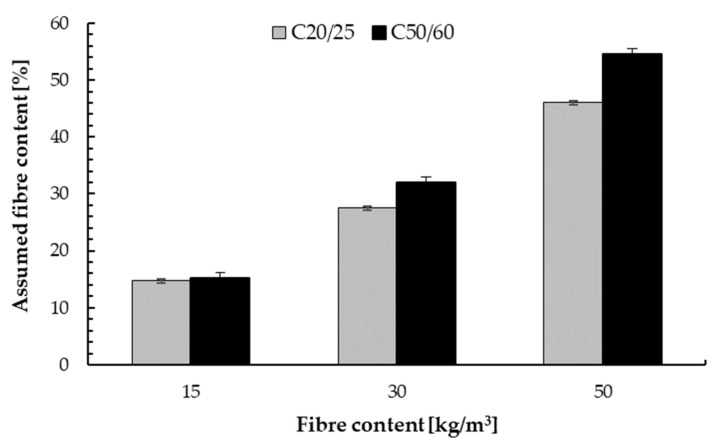
Real steel fibre content.

**Figure 8 materials-14-07757-f008:**
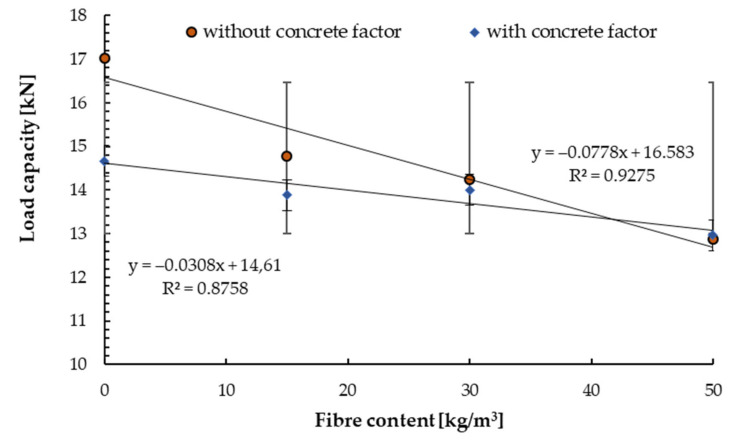
Mean load in a test with standard deviations for non−cracked concrete C20/25 for tested anchors depending on the steel fibre content.

**Figure 9 materials-14-07757-f009:**
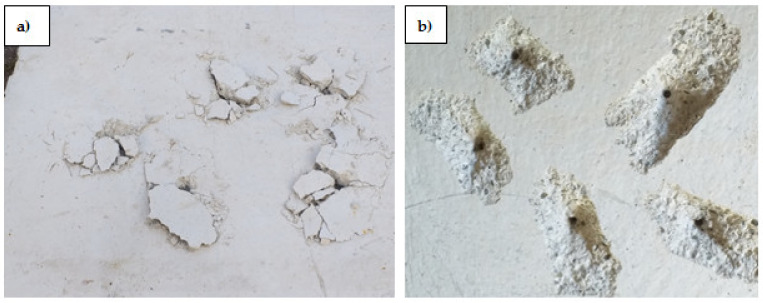
Mean load in a test for cracked concrete C20/25 for tested anchors depending on the steel fibre content. (**a**) concrete cone; (**b**) concrete cone after tests.

**Figure 10 materials-14-07757-f010:**
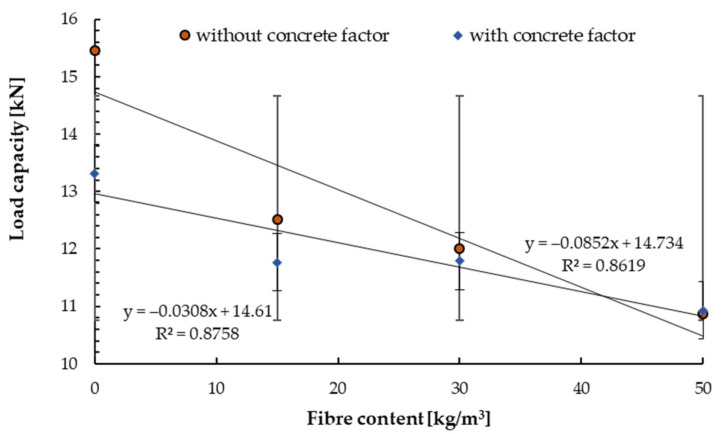
Mean load in a test with standard deviations for cracked concrete C20/25 for tested anchors depending on the steel fibre content.

**Figure 11 materials-14-07757-f011:**
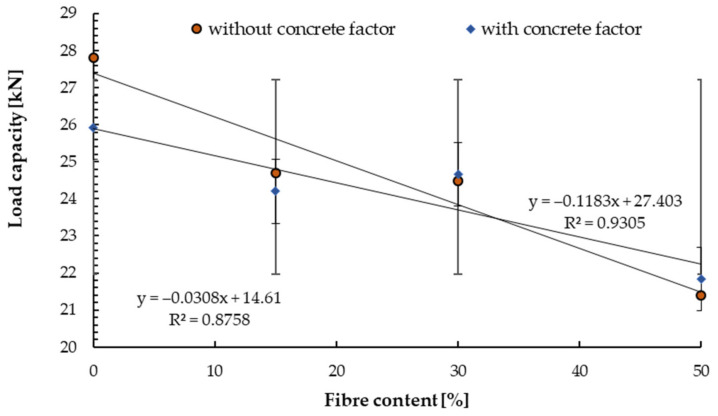
Mean load in a test with standard deviations for non−cracked concrete C50/60 for tested anchors depending on the steel fibre content.

**Figure 12 materials-14-07757-f012:**
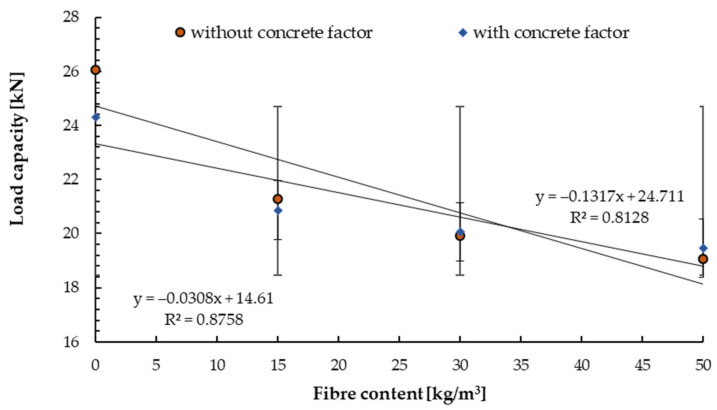
Mean load in a test with standard deviations for cracked concrete C50/60 for tested anchors depending on the steel fibre content.

**Figure 13 materials-14-07757-f013:**
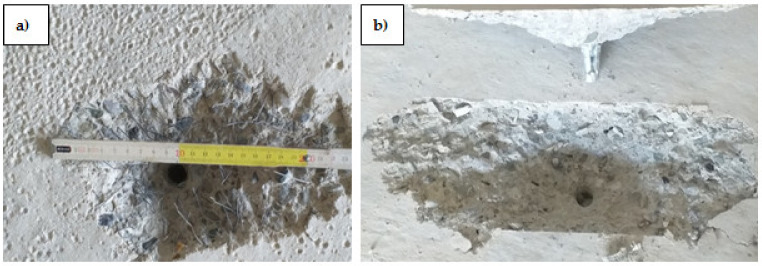
Image of destruction: (**a**) with fibre, (**b**) concrete cone.

**Table 1 materials-14-07757-t001:** Materials and installation parameters of steel expansion anchor.

Length L_w_ [mm]	Drill Diameter d_0_ [mm]	Nut Diameter d_p_ [mm]	Tensile Strength R_m_ [MPa]
140	10	17	680

**Table 2 materials-14-07757-t002:** Materials and installation parameters of steel expansion anchor.

Component-Material	Drill Depth h_1_ [mm]	Embedment Depth h_nom_ [mm]	Effective Depth h_ef_ [mm]	Nominal Torque T_inst_ [Nm]
Anchor body–carbon steel Expansion sleeve–Stainless steel A4 Hexagonal nut–Steel class 8 EN ISO 898-2 Washer–Steel DIN 125 Protection–coating (≥5 μm) acc. to EN ISO 4042	54	49	40	30

**Table 3 materials-14-07757-t003:** Chemical composition of cement.

Compositions	SiO_2_	Al_2_O_3_	Fe_2_O_3_	CaO	MgO	SO_3_	Na_2_O	K_2_O	Cl	LOI	IR
Unit (vol.%)	19.5	4.9	2.9	63.3	1.3	2.8	0.1	0.9	0.05	2.48	0.63

**Table 4 materials-14-07757-t004:** Physical properties of cement.

Properties	Specific Surface Area [m^2^/kg]	Specific Gravity [kg/m^3^]	Compressive Strength [MPa]	
Materials			After 2 Days	After 28 Days
Cement	3840	3060	28.0	58.0

**Table 5 materials-14-07757-t005:** Physical and mechanical properties of steel fibres.

Length of Fibre L [mm]	Length l [mm]	Diameter d [mm]	Height h [mm]	Tensile Strength R_m_ [MPa]
50	4.0	0.8	3.0	1100

**Table 6 materials-14-07757-t006:** Composition of 1 m^3^ concrete matrix.

Components	Concrete		
	C20/25		C50/60
		[kg/m^3^]	
CEM I 42.5 R (Górażdże, Poland)	230		420
Silica fume 2/8 (KSM, Poland)	380		464
Silica fume 8/16 (KSM, Poland)	830		645
Quartz sand 0/2 (KSM, Poland)	770		622
MasterPozzolith BV 18 C (BASF, Poland)	0.92		—
Sikament 400/30 (Sika, Poland)	1.72		—
Sika ViscoCrete-3088 M (Sika, Poland)	—		2.52
Water	140		182

## Data Availability

Not applicable.

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
