# Peer review of "Effect Steel Fibre Content on the Load-Carrying Capacity of Fibre-Reinforced Concrete Expansion Anchor"

_materials, 2021, doi:10.3390/ma14247757_

Round 1

Reviewer 1 Report

The reviewer appreciates the experimental work done by the authors. The technical contents of the paper are in general interesting. The findings from this study are useful information for field applications in the future. In the reviewer’s opinion, the goal of the work must be better explained within Abstract, Introduction and Conclusions. Moreover, the publication in the “Materials” is not recommended unless the following suggestions are taken into account within the article:

1) The current state of knowledge relating to the manuscript topic has not been covered and clearly presented, and the authors’ contributions are not emphasized. In this regard, the authors should make their effort to address these issues, by adding additional comments on the state of the art and the proposed aspects.

2) Objectives and information should be presented more clearly. Furthermore, additional comments should be added in regard to the practical value of this work, and how the industry can profit from this article.

3) Introduction: Fiber reinforced concretes can particularly improve stiffness and durability of reinforced and prestressed concrete structures. Please, mention this issue and, consequently, cite the following corresponding references:

a) Influence of prestressing on the behavior of uncracked concrete beams with a parabolic bonded tendon. Structural Engineering and Mechanics, 2021, 77(1), pp. 1–17.

4) The mechanical properties of the fiber reinforced concretes under investigation must be compared with concretes having conventional mix-design. I.e., compared with conventional concretes, the superiority of the proposed ones was not clearly explained. Please, review the corresponding parts.

5) It is unclear why the experiments have specifically been performed at 90 days, i.e., during the early-age of the fiber reinforced concretes under investigation. Long-term conditions generally affect the durability and the mechanical properties of concretes. Please, review the following references, and specify the corresponding reasons within the article:
a) RILEM draft recommendation: TC-242-MDC multi-decade creep and shrinkage of concrete: Material model and structural analysis. Model B4 for creep, drying shrinkage and autogenous shrinkage of normal and high-strength concretes with multi-decade applicability. Materials and Structures 48, 753–770.
b) Synergic influence of degrading mechanisms and induced loading by prestressing on the concrete: state of the art. Environmental Science and Pollution Research, https://doi.org/10.1007/s11356-021-17151-9.

6) The steps of the performed experiments were not presented in detail. Please, review the corresponding parts.

7) Please, provide the technical characteristics of all the equipment and devices used.

8) I suggest to the authors to edit all the text of the article with the help of a native English speaker. Grammar, punctuation, spelling, verb usage, sentence structure, conciseness, readability and writing style could be improved.

Author Response

Dear Reviewer,

Thank you very much for all your comments on our revised manuscript. Below we have provided detailed answers to your comments. All introduced corrections in the article were made in track 'changes mode'. All introduced corrections were listed and justified in the cover letter as suggested by the Editor.

We hope you will accept our new layout of the revised manuscript.

1.1. The current state of knowledge relating to the manuscript topic has not been covered and clearly presented, and the authors’ contributions are not emphasized. In this regard, the authors should make their effort to address these issues, by adding additional comments on the state of the art and the proposed aspects.

Thank you very much for your comment. We agree that the references cited in the article do not exhaust the subject of the article, especially in the field of concrete reinforced with steel fibers. The subject of steel fiber reinforced concrete is very wide and it was not the direct goal of the article, however, we tried to supplement the current state of knowledge. We also took into account the reviewer's references, for which we would like to thank you very much. We hope you will accept our revised manuscript.

1.2. Objectives and information should be presented more clearly. Furthermore, additional comments should be added in regard to the practical value of this work, and how the industry can profit from this article.

Thank you very much for your attention. Currently, fastenings with steel anchors are mainly performed in concrete foundations, and their technical assessment is based on the recommendations of EOTA and Eurocode 2. Fastenings made in fibre-reinforced concrete elements have so far been designed and implemented as for typical concrete substrates. It is not consistent with the actual working conditions of the connectors in structural fastenings. However, the rules for making structural fastenings in fibre-reinforced concrete substrates and for carrying out their technical assessment have not been developed so far. Therefore this paper aims to effect the steel fibres content on the maximum pull-out force for M10 expansion fasteners installed in fibre-reinforced concrete substrates. The article presents the pull-out strength tests carried out on M10 expansion anchors in non-cracked and cracked steel fibre reinforced concrete. Two concrete classes (C20/25 and C50/60) were tested to determine of the possibility of such fastener application in civil engineering. The obtained results are the first step to change in procedure of technical assessment for steel expansion anchors in fibre-reinforced concrete substrate and taking into account the fiber content in the designing the application of the anchors in fibre-reinforced concrete substrate. This information was added.

1.3. Introduction: Fiber reinforced concretes can particularly improve stiffness and durability of reinforced and prestressed concrete structures. Please, mention this issue and, consequently, cite the following corresponding references:

a) Influence of prestressing on the behavior of uncracked concrete beams with a parabolic bonded tendon. Structural Engineering and Mechanics, 2021, 77(1), pp. 1–17.

Thank you very much for your suggestion. Of course, we agree that this information is very important. This information was added.

1.4. The mechanical properties of the fiber reinforced concretes under investigation must be compared with concretes having conventional mix-design. I.e., compared with conventional concretes, the superiority of the proposed ones was not clearly explained. Please, review the corresponding parts.

Thank you very much for your attention. It was corrected. We hope you will be satisfied.

1.5. It is unclear why the experiments have specifically been performed at 90 days, i.e., during the early-age of the fiber reinforced concretes under investigation. Long-term conditions generally affect the durability and the mechanical properties of concretes. Please, review the following references, and specify the corresponding reasons within the article:

a) RILEM draft recommendation: TC-242-MDC multi-decade creep and shrinkage of concrete: Material model and structural analysis. Model B4 for creep, drying shrinkage and autogenous shrinkage of normal and high-strength concretes with multi-decade applicability. Materials and Structures 48, 753–770.

b) Synergic influence of degrading mechanisms and induced loading by prestressing on the concrete: state of the art. Environmental Science and Pollution Research, https://doi.org/10.1007/s11356-021-17151-9.

Thank you very much for your comment. According to the EOTA test procedure, the pull-out tests are performed not earlier than after 28 days of concrete hardening, and before the tests, the compressive strength of the substrate concrete should be determined. In fact, we know that endurance builds up over time even after 28 days, and we fully agree that 90 days is an early stage for testing. Therefore, the results presented in this paper are one of the first attempts to observe the results of the pull-out test over time, i.e. after these 90 days. We agree that long-term testing is very important in assessing the durability and the mechanical properties of concretes, however, at the present stage of testing, we do not have such results. As part of further research, we will study and analyze the increase of the load force over time for fibre-reinforced substrate. This information was added to article.

We hope you will accept our explanation and you will accept our revised manuscript.

1.6. The steps of the performed experiments were not presented in detail. Please, review the corresponding parts.

Thank you very much for your suggestion. We agree that this has not been clearly presented. We have added the steps of the performed experiments. We think that this has improved the article.

1.7. Please, provide the technical characteristics of all the equipment and devices used.

Thank you very much for your suggestion. These informations were added.

1.8. I suggest to the authors to edit all the text of the article with the help of a native English speaker. Grammar, punctuation, spelling, verb usage, sentence structure, conciseness, readability and writing style could be improved.

Thank you very much for your comment. The text was corrected according to native English suggestions. We hope you will accept our manuscript.

Once again, thank you very much for all comments and suggestions. They were very useful and helpful to us. In addition, your comments are a valuable experience for us, which we will use in our further scientific work.

Reviewer 2 Report

Thank you for the nice article.
The topic is definitely interesting and not so common. 
The general description of the material and experiments is well done.
A few things need to be improved. 

The article is missing the percentage of wire in concrete for all mixtures. The values in kg/m3 are insufficient. 

How do you explain such huge variations in Figures 8, 10, 11 and 12?
Isn't there a numerical error here? The upper and lower standard deviation is seemingly still the same - that's not right - unless you are showing the standard deviation.
The legend in figure 11 is wrong if I'm not mistaken. 

I recommend adding some interesting articles about wireframes to the introduction:
https://doi.org/10.3390/ma14226838
10.3390/ma14123235
10.3390/ma13143074

Author Response

Dear Reviewer,

Thank you very much for all your comments on our revised manuscript. Below we have provided detailed answers to your comments. All introduced corrections in the article were made in track 'changes mode'. All introduced corrections were listed and justified in the cover letter as suggested by the Editor.

We hope you will accept our new layout of the revised manuscript.

2.1. The article is missing the percentage of wire in concrete for all mixtures. The values in kg/m3 are insufficient. 

Thank you very much for your suggestion. The fiber contents were added.

2.2. How do you explain such huge variations in Figures 8, 10, 11 and 12?
Isn't there a numerical error here? The upper and lower standard deviation is seemingly still the same - that's not right - unless you are showing the standard deviation.

Thank you very much for your comment. The figures show the standard deviation. However, we agree with you that this has not been adequately highlighted in our article. So, this information was added under Figures.

2.3. The legend in figure 11 is wrong if I'm not mistaken. 

Thank you very much for your attention, but we checked and legend is correct. Figure 11 relates to non-cracked concrete C50/60, while the Figure 12 to cracked concrete C50/60. Unless you meant a pattern and linguistic errors. It has been corrected.

2.4. I recommend adding some interesting articles about wireframes to the introduction:
https://doi.org/10.3390/ma14226838
3390/ma14123235
10.3390/ma13143074

Thank you very much for your comment. These references are very interesting.We added items to our article and we believe that they improved our article. We hope you will accept our revised manuscript.

Once again, thank you very much for all comments and suggestions. They were very useful and helpful to us. In addition, your comments are a valuable experience for us, which we will use in our further scientific work.

Round 2

Reviewer 1 Report

The authors have adequately addressed my comments.

Reviewer 2 Report

Thank you,

very good edit.

Regards,